# A Long Skip Connection for Enhanced Color Selectivity in CNN Architectures

**DOI:** 10.3390/s23177582

**Published:** 2023-08-31

**Authors:** Oscar Sanchez-Cesteros, Mariano Rincon, Margarita Bachiller, Sonia Valladares-Rodriguez

**Affiliations:** 1Department of Artificial Intelligence, National University of Distance Education (UNED), 28040 Madrid, Spain; mrincon@dia.uned.es (M.R.); marga@dia.uned.es (M.B.); soniavr@dia.uned.es (S.V.-R.); 2Department of Electronics and Computing, University of Santiago de Compostela (USC), 15705 Santiago de Compostela, Spain

**Keywords:** color selectivity, skip connections, long skip connection, CNN, VGG16, Densenet121, Resnet50, feature map visualization

## Abstract

Some recent studies show that filters in convolutional neural networks (CNNs) have low color selectivity in datasets of natural scenes such as Imagenet. CNNs, bio-inspired by the visual cortex, are characterized by their hierarchical learning structure which appears to gradually transform the representation space. Inspired by the direct connection between the LGN and V4, which allows V4 to handle low-level information closer to the trichromatic input in addition to processed information that comes from V2/V3, we propose the addition of a long skip connection (LSC) between the first and last blocks of the feature extraction stage to allow deeper parts of the network to receive information from shallower layers. This type of connection improves classification accuracy by combining simple-visual and complex-abstract features to create more color-selective ones. We have applied this strategy to classic CNN architectures and quantitatively and qualitatively analyzed the improvement in accuracy while focusing on color selectivity. The results show that, in general, skip connections improve accuracy, but LSC improves it even more and enhances the color selectivity of the original CNN architectures. As a side result, we propose a new color representation procedure for organizing and filtering feature maps, making their visualization more manageable for qualitative color selectivity analysis.

## 1. Introduction

Color selectivity refers to the ability of a system to respond selectively to specific colors or wavelengths of light inside its receptive field, that is, to locate patterns based on chromatic characteristics alone or color mixed with other visual features such as texture or shape. Taylor et al. [1] showed that CNNs presented near-orthogonal color and form processing in early layers, but increasingly intermixed feature coding in higher layers. Previously, several authors [2,3,4] have shown that CNN color selectivity is low in models trained with large datasets of natural scenes such as Imagenet, which are biased towards grayish, orangish, and bluish colors. It seems clear that increasing color selectivity is necessary to take full advantage of the information in the color channels, especially in scenarios where color is a primary characteristic.

The reason for this low color selectivity may be in the architecture itself. CNNs were inspired by early findings from the study of biological vision [5]. Since Fukushima with the Neocognitron [6], who built a basic block based on V1, or Lecun et al. with the Lenet architecture [7], who continued the idea of aggregating simpler features into more complex ones with a sequential architecture of repetitive building blocks, CNNs have basically maintained the same architecture. This representational hierarchy appears to gradually transform a space-based visual into a shape-based and semantic representation [8].

However, in the last thirty years, neuroscience has evolved in the understanding of the visual area and its active role in perception, as we can see in Hubel’s later work [9] or that of Davila Teller [10], where they advanced the study of the relationship between vision and the vision system using physiological and perceptual techniques. Although the process is mainly sequential and follows the scheme “retina→LGN→V1→V2→V3→V4”, there are connections between these areas that do not follow this sequentiality. Some models have been proposed that partially skip this sequential structure [11,12,13,14] but none have been proposed to improve color selectivity.

In 1994, Ferrera et al. [15] verified that the information structure of the LGN was maintained in the macaque visual area V4, which indicates that this area, specialized in the detection of shapes and colors [16,17], processes information coming from areas V1, V2, and V3, where color is intermixed for the detection of edges, shapes, and textures, and, in addition, from the LGN, where color and other features are present but at a lower level of processing. The author hypothesized that this structure of unprocessed information from the LGN is maintained in areas V1, V2, and V3 in mammals. Recently, these direct connections between the precortical area and neocortex visual areas have been analyzed in humans, indicating that there is a direct and bidirectional connection between the LGN and V4 [18].

In this study, inspired by the direct connection LGN→V4, we propose the following hypothesis: it is possible to increase the color selectivity of a feed-forward architecture and improve its performance in classification tasks by modifying the structure of the network to reflect this direct connection. We do this by creating a long skip connection (LSC) between the output of the initial block, equivalent to the LGN, and the input of the last block of the feature extraction stage of the network so that this last block, similar to area V4, processes both branches together.

To evaluate this proposal, we selected several classic CNN architectures, VGG16, Densenet121, and Resnet50, which have sufficient blocks for the proposed connection to be functional and present different types of skip connections. Specifically, VGG16 does not have skip connections, Densenet121 uses skip connections via concatenation and Resnet50 uses skip connections via addition, which allows us to analyze the LSC extensively.

Our goal is to enhance the accuracy of current CNN models by using a long skip connection that boosts their ability to recognize colors. Our study improves existing methods for measuring color distinction in a more precise way and examines how it affects accuracy through filter removal. We also introduce a novel technique to assess color-related filters, simplifying the analysis process by organizing them based on color and selectivity. This reduces the number of filters needing examination.

Accordingly, the remainder of the paper is organized as follows. Section 2 presents preliminary background information related to studying color selectivity in CNNs and different skip connection typologies. Section 3 describes the implementation of LSC on several classic CNN architectures, and a simplified procedure to analyze quantitatively and qualitatively color selectivity. Section 4 describes the experiments conducted to demonstrate color selectivity improvements of models with LSC. Section 5 shows experimental results and discusses the results along with observations. Finally, Section 6 concludes the paper.

## 2. Related Work

### 2.1. Color Selectivity in CNNs

Few studies have analyzed color selectivity in neural networks. Usually, the problem has been addressed pragmatically, testing different color representations and choosing the one with the best performance. In the last years, color selectivity in CNNs has been studied by analyzing the activation of neurons in the network in response to particular stimuli. Building on the work of Shapley and Hawken [19] on how color is encoded in the human vision system, Rafegas et al. [3,20] proposed a method to explore how color is encoded in a trained artificial network. They defined a representation to visualize the neuron activity that they named “Neuron Feature” (NF) and it consists of an image obtained by averaging the set of N-top image crops that maximally activate the neuron. Although this representation has certain limitations since it is based on the assumption that the color properties of kernels equal the color properties of their corresponding mean image patches, this qualitative representation is a proxy for what the neuron responds to: If NF has a well-defined hue, the neuron responds only to that range of hues, whereas if it has a grayish color, it responds to a wider range of colors, i.e., its response is less color-dependent. In addition, they defined a color selectivity index (CSI) for each neuron in the network by comparing its activation in response to color and grayscale images. Engilberge et al. [2] also defined a color sensitivity index by comparing neuronal activation in color and grayscale images, but focused exclusively on quantitative metrics. Later, several methodologies for the analysis of color opponency and spatial and color tuning have been defined [4,21,22], but they use prepared datasets which are beyond the scope of our study. In all these studies, the input images provided to the models are in the RGB color system.

#### 2.1.1. Color Selectivity Analysis of Individual Neurons

According to Rafegas and Vanrell’s methodology [3,20], an “image crop” represents the image area of the receptive field that activates a neuron. For qualitative analysis, they visualized the kind of inputs that trigger the neuron’s response in the spatial-chromatic domain by means of the concept of Neuron Feature (NF), which is the weighted average of the N image crops that activate the neuron the most in the dataset (Equation (Equation 1)).
(1)$$NF\left(n_{i}\right)=\frac{1}{N_{max}}\sum_{j=1}^{N_{max}}w_{j,i}I_{j}$$
where $n_{i}$ is neuron *i* (neuron *i* that belongs to a feature map *f* of a layer *L*); $N_{max}$ is the configurable parameter that sets the number of images that most activate the neuron to be considered; and $w_{j,i}$ is the weighting factor for the image crop of image *j*, $I_{j}$. This weighting factor $w_{j,i}$ is defined as the activation value normalized to the highest activation of the neuron in the entire dataset (Equation (Equation 2)).
(2)$$w_{j,i}=\frac{a_{j,i}}{a_{max,i}}$$
where $a_{j,i}$ is the value of the activation function obtained by $n_{i}$ in $I_{j}$, and $a_{max,i}=max\left(a_{k,i}\right)\forall k$ is the highest value of activation function of $n_{i}$ in the entire dataset.

For quantitative analysis, they defined the color selectivity index (CSI), which was obtained by comparing the value of $w_{j,i}$ of the original color image with the value $w_{j,i}^{\prime }$ of the same image in grayscale (Equation (Equation 3)). CSI values are in the interval [0, 1], with the extreme cases CSI = 0 (no color selectivity) when $w_{j,i}$ and $w_{j,i}^{\prime }$ are equal, and CSI = 1 (maximum color selectivity) when $w_{j,i}^{\prime }$ is 0.
(3)$$CSI\left(n_{i}\right)=1-\frac{\sum_{j=1}^{N_{max}}w_{j,i}^{\prime }}{\sum_{j=1}^{N_{max}}w_{j,i}}$$

Figure 1 shows five NF from the last layer of the fifth convolution block with low, medium and high CSI values. We choose neurons at this level to show the relationship between CSI and NF because this is where we will further analyze the benefits of the proposed architecture. When CSI is low, the image crops that activate the neuron the most (Figure 1b) have heterogeneous hue ranges (yellows, whites, greens, blues, etc.), whereas when CSI is high, the images have highly localized hue ranges. This is also visible in the color hue distribution of the pixels in the top 100 images of Figure 1c, which shows that it is unimodal and narrow when the CSI is high, whereas it is multimodal and flat when the CSI is low.

### 2.2. Skip Connections

Skip connections are a type of shortcut that connects the output of one layer to the input of another layer that is not adjacent to it. Srivastava et al. [11] introduced the concept of skip connections with their Highway Networks, which are characterized by the use of gating units which learn to regulate the flow of information between layers. However, skip connections achieved fame with He et al.’s work on Residual Networks (ResNet) [12], which, although a particular case of the previous ones, was the foundation of the first-place winning entries in all five main tracks of the ImageNet and COCO 2015 competitions, which covered image classification, object detection and semantic segmentation. These networks use skip connections via addition to learning residual functions with reference to the layer inputs. They allowed to solve the degradation problem, which facilitates the design of deeper neural networks.

Huang et al. [13] proposed another architecture that also uses skip connections: Densenet. In this case, Densenet uses skip connection via concatenation of each layer with the following ones to encourage feature reuse and also to alleviate the vanishing-gradient problem. However, an essential part of CNNs is the down-sampling of layers, which reduces the size of feature maps through dimensionality reduction to achieve higher computation speeds. To enable this, Densenets are divided into Dense Blocks sequentially connected, with no skip connections between them.

Another architecture that uses skip connections is U-net (Ronneberger et al. [14]), which is a symmetrical encoder-decoder architecture in which each decoder layer receives the output of the same layer of the encoder concatenated with the input of the previous layer of the decoder. This architecture was used to learn fine-grained details, making it ideal for semantic segmentation.

There are other popular architectures, such as Inception [23] or Xception [24], that we have not analyzed; however, the reader can note that skip connections have always been used within the layers of a block or between symmetrical layers of an encoder-decoder architecture, but not between blocks in classification tasks, and not for the purpose of enhancing color selectivity.

## 3. Methods

In this section, we describe the implementation of LSC on the VGG16, Densenet121 and Resnet50 architectures and introduce two new procedures for analyzing color selectivity to show the improvements it introduces. To simplify the notation, we will refer to a generic architecture with *M* (Model) accompanied by the subscripts *O* when referring to the original architecture ($M_{O}$) and with LS when referring to the architecture with LSC ($M_{LS}$). In addition, we will use VGG to refer to VGG16, Densenet for Densenet121 and Resnet for Resnet50. Therefore, for example, to refer specifically to the VGG16 architecture modified with LSC, we will use the notation $VGG_{LS}$.

### 3.1. The Proposed LSC Architecture

As mentioned in the introduction, we propose establishing an LSC between the output of the first block and the input of the last block of the feature extraction stage of the network. Our proposal was tested on three classic CNN architectures with and without skip connections: VGG, Densenet and Resnet. All of these architectures use a similar down-sampling strategy along the network. However, there were differences between‘them.

We numbered the blocks of the feature extraction stage according to the pooling operations: 112 × 112 (first block), 56 × 56 (second block), 28 × 28 (third block), 14 × 14 (fourth block) and 7 × 7 (fifth block). The classification stage consists of one or more dense layers. Figure 2 shows the structure of both architectures: the original, $M_{O}$, and the one with LSC, $M_{LS}$. Figure 2c shows the receptive field of a neuron in the fifth block of $M_{LS}$. In the fifth block, the receptive field is composed (by adding or concatenation depending on the specific model where the LSC is implemented) of the fourth block output (forward path) and the LSC that originates from the first block. The LSC reduces the dimensions of the first block output by max-pooling, allowing the composition.

Table 1, Table 2 and Table 3 list the details of the implementation of the three architectures. The images are in RGB, and the input layer of all models consists of three channels. In VGG, the architecture is based on similar blocks (convolution layers plus an output max-pooling layer). In Densenet and Resnet, there is a convolution layer at the input and then five and four functional blocks respectively. While Resnet and VGG maintain the same dimensionality reduction pattern in the last block (the last block input is 14 × 14 and its output is 7 × 7), in Densenet, the last block input is 7 × 7 and its output is also 7 × 7. We now examine each model in detail.

The VGG architecture consists of five blocks in the feature extraction stage, and two fully connected layers and one softmax layer in the classification stage (Table 1). Each block has several convolution layers and a max-pooling layer that halves the size of the feature maps. The number of filters increases from 64 in the first block to 512 in the last block. The LSC in $VGG_{LS}$ is established between the output of the first block (112 × 112) and the fourth (14 × 14) using three 2 × 2 max-pooling operations. Therefore, the fifth block receives the concatenated output of the fourth block and the down-sampled output of the first block. The classification stage has two dense layers of 4016 units and a soft-max layer that depends on the number of classes in each dataset.

The Densenet architecture is composed of “dense blocks” followed by a “transition layer”, which halves the size of the feature maps (see Table 2). Each dense block has three convolution layers: 1 × 1, 3 × 3, and 1 × 1. The number of filters in each layer varies as the network advances in depth from 64 to 2048, as well as the number of repetitions, which are concatenated. The transition layer is composed of a 1 × 1 convolution layer and a 2 × 2 and stride = 2 average pooling layer. The first block comprises the first convolution layer and the max-pooling layer, which reduces the dimension to 56 × 56. In the fifth block, unlike in VGG, where the dimension is reduced from 14 × 14 to 7 × 7, the fifth block has an input of 7 × 7 and no reduction. Therefore, the LSC from the first block reduces the dimension to 7 × 7. The classification stage has a soft-max layer that depends on the number of classes in each dataset.

The Resnet architecture is composed of “resnet blocks” containing three convolution layers: 1 × 1, 3 × 3, and 1 × 1, with a max-pooling of 2 × 2 and stride = 2 at the end to reduce the dimension. The output block has a 7 × 7 average-pooling and a soft-max layer that depends on the number of classes in each dataset. The LSC is established between the output of the first block and the output of the fourth block, performing an adding operation to maintain the type of skip connections used in this architecture. Owing to this addition operation, we increase the number of filters of the output of the first block from 64 to 1024 using a 1 × 1 convolution layer.

### 3.2. Evaluation of Color Selectivity

To demonstrate that the improvements introduced by the LSC are related to color processing, we performed two types of analysis: a quantitative analysis of the filter distribution according to color selectivity and a qualitative analysis of the filter’s response to color hue.

As mentioned above, we focused our analysis on the fifth block, as this is where the information coming from the different layers and from the LSC connection is combined to create the final features to be used in the classification stage.

#### 3.2.1. Color Selectivity Properties of Filters

To analyze color selectivity, we will use the method of Rafegas and Vanrell described in Section 2.1.1 to obtain the color selectivity index (CSI) and the neuronal feature image (NF) of each neuron. However, these values are oriented towards a neuron-level analysis. As it is impractical to analyze the response of all neurons individually, we simplified the analysis by selecting a representative neuron of each layer, reducing the number of elements to manage.

We evaluated the CSI values in the neurons of each feature map of the fifth block output and found that their variance followed a decreasing exponential distribution ($a^{bx}$; a = 109.76; b = −73.60), which is very narrow and indicates that any neuron in the feature map can be representative of the filter behavior. Therefore, for each feature map, we selected an active and centered neuron to obtain the CSI and NF values of the filters. The CSI difference of this representative neuron with respect to the average of the feature map is 0.07(0.000025SD), which we consider insignificant for this study.

#### 3.2.2. Qualitative Color Selectivity Analysis

A feature map shows the filter response to an input image. In order to analyze differences between $M_{O}$ and $M_{LS}$ responses, we can compare all the feature maps of each block output. However, it is very complicated and cumbersome when the block has many feature maps because it is necessary to visualize all these monochrome images together (512, 768, and 2048 feature maps in VGG, Densenet, and Resnet, respectively).

To facilitate the analysis of color selectivity for a particular image, we propose the following color representation of the feature maps in the HSV space: H (hue) is the average hue of the NF of the filter that generates the feature map, S (saturation) is the CSI value of the filter, and V (value) is the normalized activation value $w_{j,i}$ for image *j*. With this representation, we can reorganize the feature maps according to the CSI intervals and select the most active filters for each one. This reduces the number of images displayed, which remarkably simplifies the analysis and, as will be seen later, does not influence the results.

In Figure 3, we show an example of how the new representation facilitates the analysis of the feature maps at the fifth block output in $VGG_{LS}$ (we just show a sample of 100 out of 512 feature maps for the sake of clarity). Given the input image shown in Figure 3a, Figure 3b shows the feature maps ordered according to their positions in the architecture, which is difficult to follow. Figure 3c is an intermediate representation in the HSV space that groups feature maps in 12 hue ranges. The light gray boxes represent hue ranges without feature maps. In this representation, the maps are sorted by their hue range and intensity (average activation), facilitating the visualization of the visuospatial features of the original image they capture, such as contours, shapes, textures, and specific areas. Finally, to further simplify the analysis, the highest intensity feature maps for each hue range and CSI interval are selected (Figure 3d).

## 4. Experimental Setup

### 4.1. Datasets

Table 4 and Figure 4 summarize the characteristics of the four datasets used in our experiments: Imagenette [25], Tiny Imagenet [26], Birds [27] and Flowers [28]. We chose these datasets because of their varied themes (natural scenes, people, artifacts, animals, flowers, etc.) and different color characteristics, with images containing monochrome and polychrome elements, with high and low saturation, and with a variable number of classes (from 10 to 315) and images (between 5000 and 100,000). Furthermore, there is an interest in the community to improve the performance of models with small datasets [29], since in the industry it is complex to obtain enough examples with homogeneous quantities to represent each class. In this scenario, where it is necessary to improve the feature extraction capability, the comparison of the proposed improvement will be more effective than using large datasets. Imagenette and Tiny Imagenet are two datasets derived from Imagenet [30]. The objects of interest appear in different sizes and positions, complete in the foreground or segmented and occluded by other objects. Birds and Flowers are two specialized datasets, one on birds and the other on flowers. The first one has the objects of interest in the foreground, and the second one in the foreground and background. All four datasets contain mainly natural scenes, so their color hue distribution shows a high concentration of pixels in the orange range (due to the presence of organic objects) and somewhat less in the bluish range (mainly due to the presence of sky or water regions in the background). Only the Flowers dataset contained no peak in the bluish range.

### 4.2. Setup

Because our purpose is to analyze the performance improvement of the new architecture with respect to color selectivity, we used the default configuration parameters offered by Keras to train the models, without any technique to improve accuracy (e.g., preprocessing, batch normalization, etc.). We used a learning rate lr = 0.00001 in VGG to prevent dead neurons, and lr = 0.0001 in Densenet and Resnet; batch size = 32; Adam optimizer; and cross-entropy as the loss function. We selected the model with the highest validation accuracy in each training session (usually obtained in approximately 15 epochs), and then chose the one with the best performance in five training sessions. Table 5 presents the count of trainable parameters for the models based on the number of classes. For VGG16, in its LSC variant, the trainable parameters in the fifth block’s input increase by 294,912. This corresponds to a percentage increase of 0.228% for 10 classes, 0.226% for 200 classes, 0.225% for 315 classes and 0.227% for 102 classes. Moving to Densenet 121, its LSC variant leads to an increment of 74,208 trainable parameters for 10 classes, representing an increase of 1.92%. Similarly, for 200 classes, there is an increase of 83,328 parameters, reflecting a 2.07% rise. For 315 classes, the increase is 88,848 parameters, resulting in a 2.16% augmentation. Lastly, for 102 classes, an increase of 78,624 parameters, or 1.99%, is observed. Shifting to Resnet50, its LSC variant results in an enlargement of 590,848 trainable parameters in the input of the fifth block. This change corresponds to a 2.50% increase for 10 classes, 2.17% for 200 classes, 2.02% for 315 classes and 2.33% for 102 classes.

### 4.3. Evaluation

A quantitative and qualitative analysis of the effect of color selectivity on the performance of the different architectures was carried out on the fifth block output, where the impact of LSC appears. This is the last block of the feature extraction stage, where the information from the forward path and the LSC is merged, and the final visuospatial features used in the classification stage are generated.

First, we compare the global accuracy of $M_{O}$ and $M_{LS}$ architectures. Next, to understand the effect on performance of filters of different CSI intervals, we conduct an ablation study where we “turn off” filters based on their CSI. Finally, we used the method proposed in subsection Methods.B to evaluate the model response to color hues.

#### 4.3.1. Performance Analysis

We followed the methodology described in the subsection Related Work.A to obtain the NF and CSI values of every filter of the fifth block output based on a subset of 3000 images from each validation dataset, except for Flowers, where there are only 1500 images. The 100 images that most activated the selected neuron of each filter were used. We assessed whether there was any improvement in performance with $M_{LS}$ by calculating the difference in accuracy between $M_{O}$ and $M_{LS}$.

#### 4.3.2. Filter CSI Analysis

We conducted an ablation study of the filters of the fifth block output as a function of their CSI to evaluate their impact on classification accuracy. Although ablation techniques have been used in CNNs [31], there have been no previous studies on color selectivity. We checked several CSI interval sizes to group filters. Initially, we tested a small size CSI interval (bin size = 0.1, ten intervals) and detected no significant changes. Therefore, we increased the interval size (bin size = 0.25) and obtained four CSI intervals: low (L) = [0, 0.25], medium-low (M-L) = [0.25, 0.50], medium-high (M-H)= [0.50, 0.75], and high (H) = [0.75, 1]. Significant differences were observed in this case.

The procedure to analyze the different architectures for each CSI interval first calculates the effect of ablation on the model accuracy, $DA_{M_{X},csi}$ (Equation (Equation 4)), and then the difference between this effect in the $M_{O}$ and $M_{LS}$ variants, $DM_{M,csi}$ (Equation (Equation 5)).
(4)$$DA_{M_{X},csi}=Acc\left(M_{X}\right)-Acc\left(M_{X,A_{csi}}\right)$$
where $M_{X}$ in $\left\{M_{O},M_{LS}\right\}$ represents a particular architecture; csi∈{L,M−L,M−H,H} is the CSI interval; $A_{csi}$ represents the ablation of the filters on CSI interval csi; $Acc(M_{X})$ is the accuracy without ablation; and $Acc(M_{X,A_{csi}})$ is the accuracy after ablation of filters in CSI interval csi.
(5)$$DM_{M,csi}=DA_{M_{LS},csi}-DA_{M_{O},csi}$$
where M∈{VGG,Dense,Resnet} and csi∈{L,M−L,M−H,H}. $DA_{M_{X},csi}$ indicates the contribution of the feature maps of a particular CSI interval to the final performance and $DM_{M,csi}$ indicates whether the ablation affects $M_{O}$ or $M_{LS}$ more. Therefore, the $M_{LS}$ variant improves color selectivity if the difference $DM_{M,csi}$ is positive and larger in higher CSI intervals, where the filters are more color selective.

#### 4.3.3. Color Hue Analysis

We conducted an experiment to compare the responses of the fifth block output filters to different hues. Instead of using a monochrome image, which would provoke an unknown filter response because it is outside the dataset population, we used an image of the “green broadbill” in the Birds dataset in which hue is one of the most representative features of the bird and whose shape-color combination is not very present in the dataset. We created two synthetic sets of images with different hues (red, yellow, green, cyan, blue, and magenta), one with texture details and the other with only the silhouette, as in the work of Taylor et al. [1]. We use both sets of images to determine whether the figure is detected solely based on its contours or whether its texture, including different shades of color, also plays a role. Hue is a significant color characteristic, and this experiment will evaluate whether there is an improvement in selectivity concerning hue, both in monochromatic signals (silhouettes) and in those with intensity variation (texture), and in what hue ranges (those very present or not so present in the dataset).

#### 4.3.4. Qualitative Analysis of Filter Response

Finally, to complete the comparative study, we used the color representation of the feature maps of the fifth block output to qualitatively characterize the differences in the types of features to which the filters respond. To achieve this, we analyzed the feature maps of several randomly selected images from each dataset.

## 5. Experimental Results

### 5.1. Global Performance Analysis

Table 6 lists the accuracy of each model for the validation datasets. Analyzing accuracy, we draw attention to the following results:All $M_{LS}$ variants improved accuracy compared to $M_{O}$.The accuracy achieved in each dataset varied significantly: [64–75]% in Imagenette, [20–29]% in Tiny Imagenet, [47–77]% in Birds and [25–57]% in Flowers.Among the architectures, Densenet achieved the highest accuracy in almost all datasets, with few differences with respect to Resnet. VGG performed worse, especially in Birds and Flowers.$VGG_{LS}$ achieved accuracy values similar to the original $Densenet_{O}$ and $Resnet_{O}$.

Regarding the differences in accuracy between $M_{O}$ and $M_{LS}$, we highlight that:The highest differences were produced in Birds and Flowers, independently of the architecture.VGG had the highest relative difference in all datasets (42% trained on Flowers stands out).Densenet had the lowest relative difference (3%) on Tiny Imagenet.

In brief, LSC improved the accuracy of the architecture regardless of whether the original architecture already had skip connections.

The computation time during the training of LSC variants was higher, although it varies depending on the model and the number of classes in the dataset (see Table 7). For VGG16, the LSC increment is 2.49% for Imagenette, 1.29% for Tiny Imagenet, 2.70% for Birds, and 3.44% for Flowers. Regarding Densenet, the increments are 1.21% in Imagenette, 3.18% in Tiny Imagenet, 2.92% in Birds, and 4.55% in Flowers. In the case of Resnet50, the increments are 3.11% in Imagenette, 1.23% in Tiny Imagenet, 4.07% in Birds, and 4.86% in Flowers. It is worth noting that these time variations are contingent on the model and dataset; however, in no scenario does the time increase surpass 5%.

### 5.2. Filter CSI Analysis

Figure 5 shows the distribution of the fifth block output filters per CSI interval. We highlight that the distribution of filters per CSI interval varied with the dataset and was quite similar for all the architectures.

Figure 6 shows the reduction in accuracy resulting from filter ablation in each of the four CSI intervals. Analyzing, we highlight the following:In $M_{O}$, the decrease in performance tended to be greater at high CSI intervals. In $VGG_{O}$ and $Densenet_{O}$, it varied between −1% and 21%, whereas in $Resnet_{O}$, it did not exceeded 2%. Of note were the M-L interval in $VGG_{O}$ for Imagenette (21%) and the M-H and H intervals in $Densenet_{O}$ for Tiny Imagenet (14% and 16%, respectively). There were three small negative values.In $M_{LS}$, the performance decrease also tended to be greater at higher CSI intervals. The decrease varied between 1% and 17% in $VGG_{LS}$, between 0% and 26% in $Densenet_{LS}$, and between 0% and 12% in $Resnet_{LS}$. Negative values were not observed.

The difference between the effects on the accuracy of filter ablation per CSI interval, $DM_{M,csi}$, is shown in Figure 7. The H intervals in VGG for Flowers (9%), in Densenet for Imagenette and Birds (9%), and in Resnet for Tiny Imagenet (10%) stand out. Also noteworthy are the negative values in the M-L intervals of VGG and Densenet for Imagenette (−5%). The positive trend of $DM_{M,csi}$ as CSI increases is clearly shown in Figure 8, but the correlation analysis between the accuracy variation and the number of filters per CSI shows that the correlation is low in all cases (Table 8).

In summary, the models increased the filters in the L (or even M-L) interval; however, the highest contribution to accuracy occurred in the H interval, supporting our hypothesis that LSC increases color selectivity.

### 5.3. Color Hue Analysis

Table 9 shows the Pearson correlation, for each dataset, between the hue distribution of pixels in the dataset and the number of filters in the model trained with the same dataset. It is above 0.60 in three out of four datasets (Imagenette, Tiny Imagenette and Birds); however, it is low in Flowers (between 0.17 and 0.32). In this case, the yellow-green range is the most prevalent in the distribution of pixels in the dataset, whereas the orange-yellow range contains more filters. This difference can be explained by the fact that the former is more prevalent in the background and the latter in the figure.

Figure 9, Figure 10 and Figure 11 show the most activated feature maps for each CSI interval for the silhouette and texture datasets. We highlight the following results:There were no significant differences between the maps of the silhouette and texture datasets.The models had maps with more defined silhouettes in the H interval.The difference between models was minimal in the L, M-L, and M-H intervals, except for $VGG_{LS}$ and $Resnet_{LS}$, which had maps with active areas in all hue ranges.In the H interval, all $M_{LS}$ variants had feature maps in more hue ranges than their $M_{O}$ counterparts, especially in the red-magenta-blue range; in particular, $VGG_{LS}$ achieved higher activation.In addition, in the H interval, $VGG_{LS}$ had maps in all hue ranges of the dataset, whereas $VGG_{O}$ only had less active maps in the red and orange ranges. $Densenet_{O}$ and $Densenet_{LS}$ had maps in the most frequent ranges of the Birds dataset but with incomplete silhouettes except on the orange hue, where $Densenet_{LS}$ obtained a complete figure. $Resnet_{O}$ has maps in all dataset hue ranges, like $VGG_{LS}$, and $Resnet_{LS}$, except for the magenta to red range.

In summary, the models with LSC improved the response of the filters to the hue ranges present in the dataset for both silhouettes and textures. The difference was higher in the architecture without skip connections. On the other hand, the ability of models to detect less present hue ranges varies. $VGG_{LS}$ is the model with the strongest ability to detect both strongly and weakly present nuances. $Resnet_{O}$ and $Resnet_{LS}$ also detected them, but at a lower activation level.

### 5.4. Qualitative Analysis of Filter Response

Figure 12 shows the color representation of the feature maps of the fifth block output for four randomly selected images, one from each dataset. In the visual analysis, we highlight the following points:The image from Imagenette (Figure 12a) has a strong contrast yellow-cyan between the figure and the background. The hue histogram shows that the cyan range is the most present in the image. Both variants of VGG get maps detecting the figure in yellow and the background in cyan; however, $VGG_{LS}$ has maps of the orange areas in addition to the yellow parachute areas. $Densenet_{O}$ only has maps with very specific edges of the parachute or the silhouette in the low CSI intervals, whereas $Densenet_{LS}$ detects the silhouette of the figure in the M-H interval of the red range, although the activation area is larger than the corresponding figure in the image. $Resnet_{O}$ detects the background but not the figure, and $Resnet_{LS}$ detects both the figure and the background.The image from Tiny Imagenet (Figure 12b) shows less contrast between the background and the figure. The hue histogram shows that the yellow-green range is the most present. $VGG_{O}$ detects textures of the background in orange and yellow hues, while $VGG_{LS}$ detects textures of both the figure and background areas in green ranges as well. $Densenet_{O}$ and $Densenet_{LS}$ only have maps with very small areas of the figure in the predominant hues in the image: red, orange, yellow, or green; less in $Densenet_{O}$ than in $Densenet_{LS}$. $Resnet_{O}$ does not detect areas of the figure and $Resnet_{LS}$ does, even the silhouette in the orange or yellow ranges.The image from Birds (Figure 12c) shows a hue histogram where the yellow and cyan ranges are the highest, and green has a lower presence, representing areas in the background. $VGG_{O}$ and $VGG_{LS}$ detect the silhouettes of the figure in their yellow hues. In addition, $VGG_{LS}$ detects the texture. $Densenet_{O}$ only detects reduced areas of the figure, whereas $Densenet_{LS}$ gets larger zones, particularly in the figure. $Resnet_{O}$ has maps detecting background textures and $Resnet_{LS}$ also detects the silhouette and background areas but with less definition than $VGG_{LS}$.Finally, the image from Flowers (Figure 12d) shows a hue histogram with the yellow-green range as the most relevant, followed by orange. $VGG_{LS}$ detects the full silhouettes in the orange-yellow ranges and $VGG_{O}$ either the edges or only one of the flowers. $Densenet_{O}$, $Densenet_{LS}$, and $Resnet_{O}$ detect reduced areas of the figure in the orange-yellow ranges. $Resnet_{LS}$, like $VGG_{LS}$, detects the full silhouettes but with less detail.

In cases with a higher contrast between the figure and background, such as in the Imagenette parachute (cyan range in the background and orange range in the figure), the improvements from LSC are smaller. However, in cases with less contrast (such as the Tiny Imagenet case or the Flowers case), LSC detects more features in the figure. This effect is higher for VGG and ResNet than for Densenet. This shows that LSC improves color selectivity in complex patterns, where contrast is more difficult to achieve, and facilitates feature extraction.

## 6. Conclusions and Future Work

In this paper, we focused on enhancing the color selectivity of CNN architectures. Bio-inspired by the direct connection between the LGN and area V4, we presented an LSC architecture that connects the first and last blocks of the feature extraction stage, which allows the incorporation of low-level features, detected near the RGB input, to the definition of the more abstract features used in the classification stage.

It has been demonstrated that our proposal improves the performance of the original CNN architectures by enhancing color selectivity, regardless of whether they have skip connections. On the one hand, this improvement correlates with CSI (the higher the CSI interval, the higher the improvement). On the other, the size of the improvement is more prominent in datasets where color is a more significant feature but the particular CSI redistribution of filters depends exclusively on the characteristics of the training dataset.

However, not all improvements were due to color processing. Ablation studies show that, although high CSI filters produce the most significant improvements, there are often improvements due to low CSI filters. Therefore, although we focused on the analysis of color selectivity in this study, the LSC connection probably also improves the treatment of low saturated colors and achromatic textures. This point should be studied experimentally using more controlled datasets.

Hue analysis was facilitated by the proposed qualitative methodology, which allows us to group and sort the feature maps. Although the hue distribution of filters correlates with that of pixels in the dataset, the LSC connection enables the setting of selective filters for color hues that are poorly represented in the dataset.

To conclude, in this study, we analyzed how the filters of the last block of the feature extraction stage improved their color selectivity when using LSC. In future studies, it would be interesting to establish the relationship between color selectivity in the feature extraction and classification stages, especially by analyzing the role of color selectivity in per-class accuracy. On the other hand, the study of the ablation of filters by CSI and hue could be used to improve performance and reduce computational costs.

## Figures and Tables

**Figure 1 sensors-23-07582-f001:**
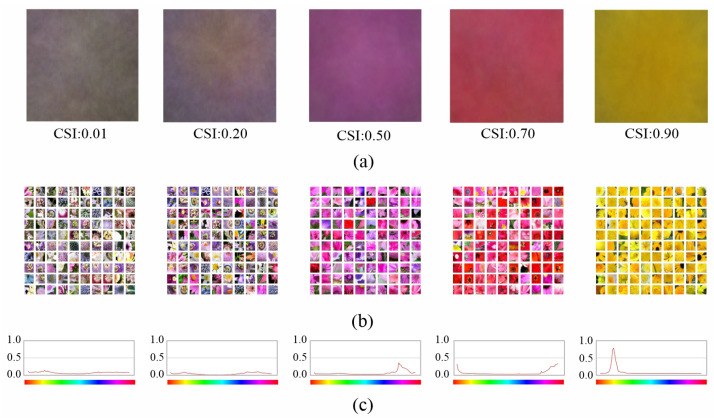
Examples of NF associated to neurons of the last layer of the fifth convolution block with different CSI values: (**a**) NF images; (**b**) image crops of the top 100 images that activate the neuron the most; and (**c**) color hue distribution of the 100 image crops.

**Figure 2 sensors-23-07582-f002:**
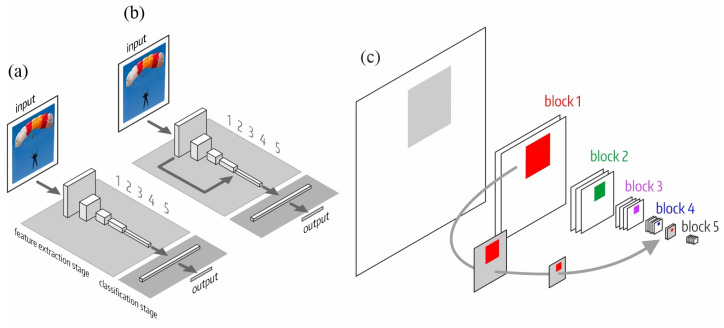
LSC architecture inspired by the connection between LGN and V4. (**a**) Original architecture ($M_{O}$); (**b**) new LSC architecture ($M_{LS}$); and (**c**) detail of the receptive field of a neuron of the fifth block, which results from the composition of the fourth block output (forward path) and the LSC that comes from the first block.

**Figure 3 sensors-23-07582-f003:**
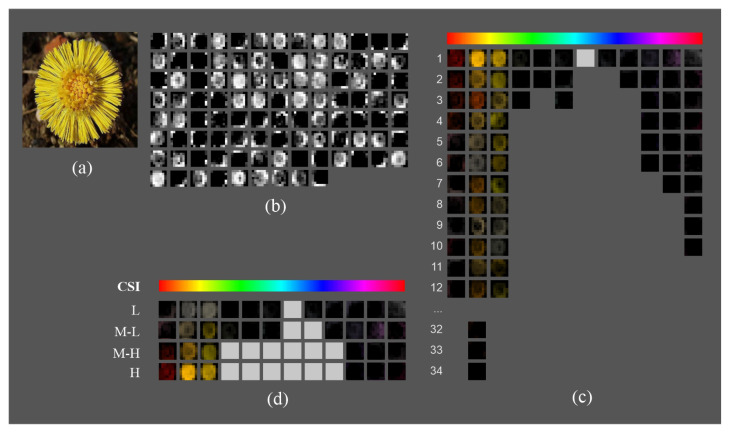
Color representation of the feature maps of the fifth block output layer to facilitate color selectivity analysis: (**a**) input image; (**b**) grayscale representation of the sample of 100 out 512 feature maps in the fifth block output layer in $VGG_{LS}$ architecture; (**c**) HSV representation of the feature maps grouped by hue ranges and sorted by average activation value; and (**d**) simplified visualization consisting of the feature map with the highest average activation per hue range and CSI interval.

**Figure 4 sensors-23-07582-f004:**
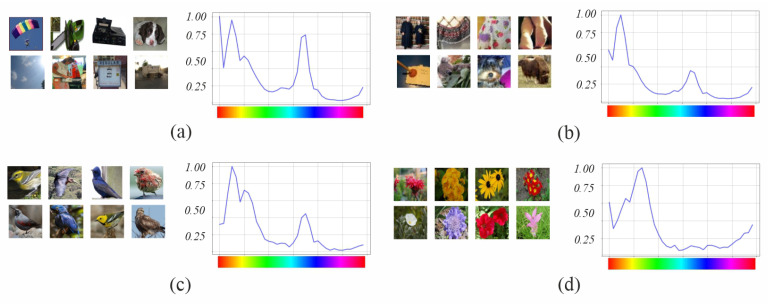
Overview of the four datasets used to evaluate the models: (**a**) Imagenette; (**b**) Tiny Imagenet; (**c**) Birds 315 species; and (**d**) 102 category flower.

**Figure 5 sensors-23-07582-f005:**
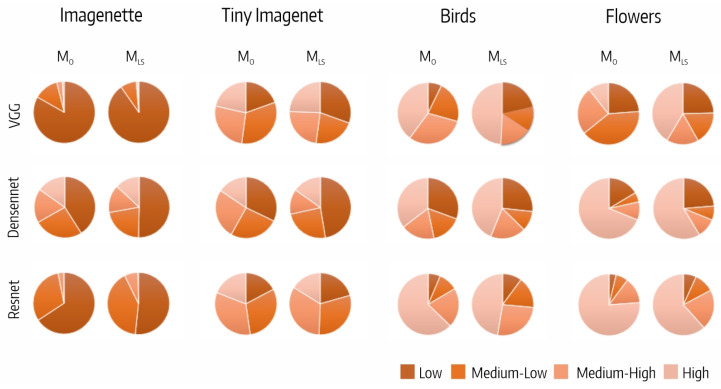
Distribution of fifth block output filters per CSI interval.

**Figure 6 sensors-23-07582-f006:**
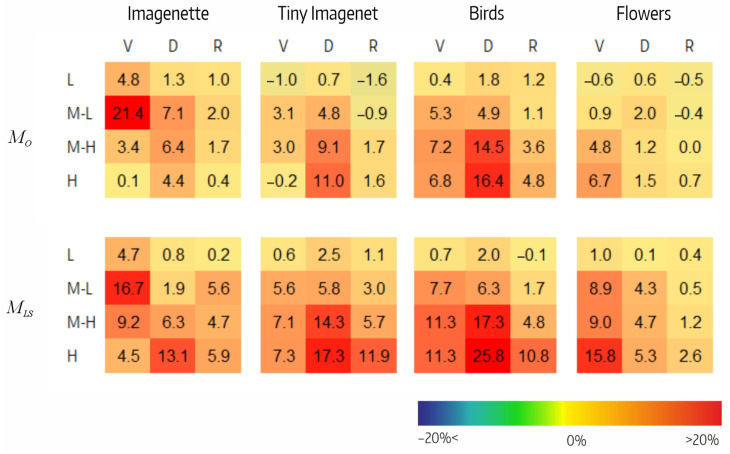
Effect of ablation in $M_{O,csi}$ and $M_{LS,csi}$ variants per CSI interval ($DA_{M_{X},csi}$) for VGG (V), Densenet (D), and Resnet (R) architectures on the four datasets.

**Figure 7 sensors-23-07582-f007:**
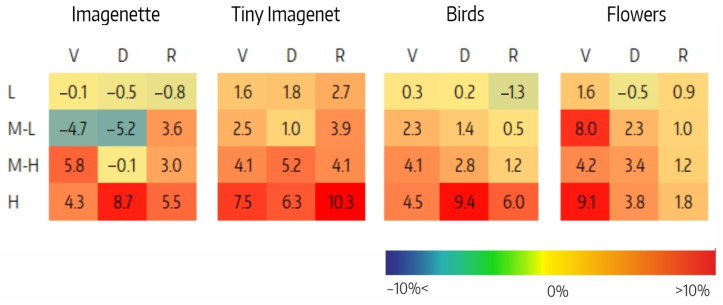
Difference between the effects of ablation in the $M_{O,csi}$ and $M_{LS,csi}$ variants per CSI interval ($DM_{M,csi}$) for VGG (V), Densenet (D), and Resnet (R) architectures on the four datasets.

**Figure 8 sensors-23-07582-f008:**
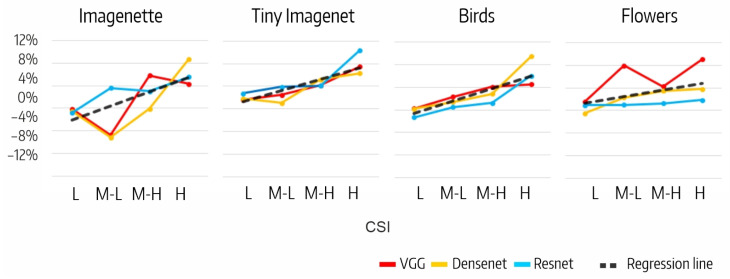
Relationship between $DM_{M,csi}$ and CSI interval.

**Figure 9 sensors-23-07582-f009:**
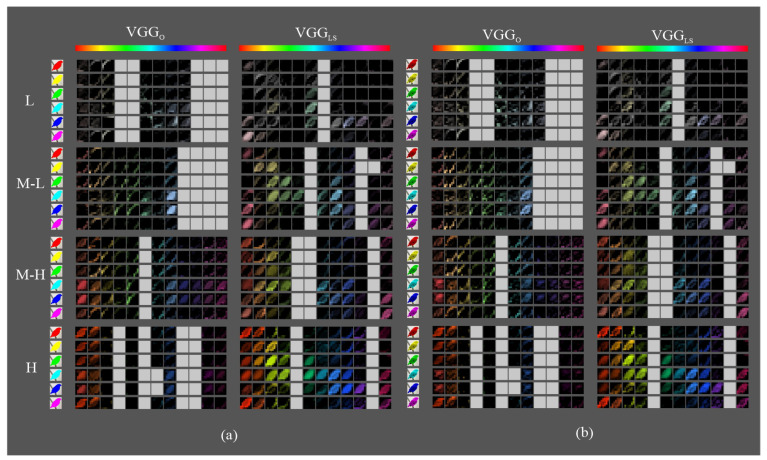
Fifth block filters response in VGG variants to images with objects of interest of different hues: (**a**) only silhouette; (**b**) with texture details.

**Figure 10 sensors-23-07582-f010:**
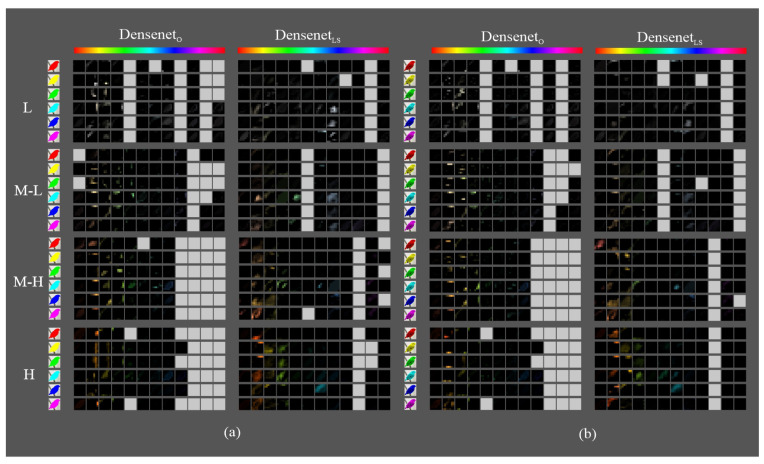
Fifth block filters response in Densenet variants to images with objects of interest of different hues: (**a**) only silhouette; (**b**) with texture details.

**Figure 11 sensors-23-07582-f011:**
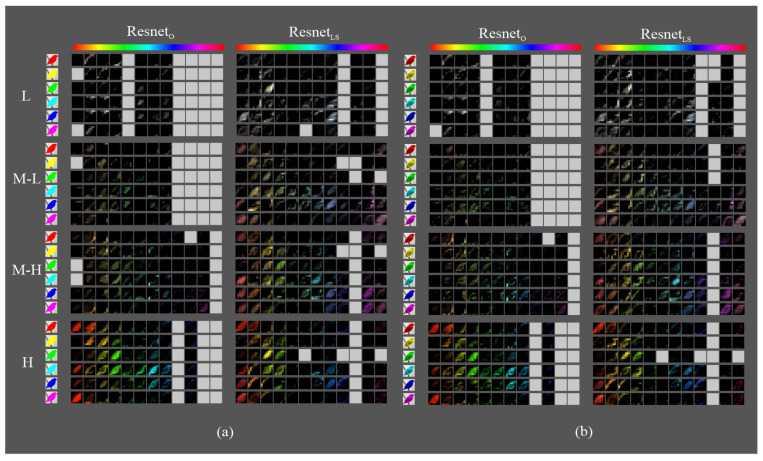
Fifth block filters response in Resnet variants to images with objects of interest of different hues: (**a**) only silhouette; (**b**) with texture details.

**Figure 12 sensors-23-07582-f012:**
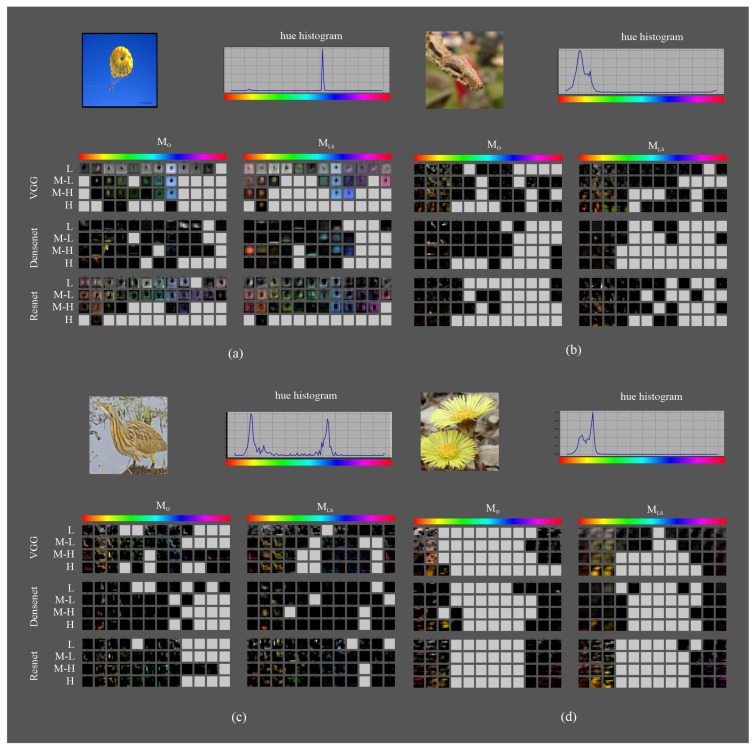
Randomly selected images in each dataset as an example for the feature map visualization according to the proposed methodology: (**a**) Imagenette; (**b**) Tiny Imagenet; (**c**) Birds 315 species; and (**d**) 102 category flower.

**Table 1 sensors-23-07582-t001:** VGG and $VGG_{LS}$ architectures. The blocks are composed of several convolution layers and a max-pooling layer at the output. The max-pooling is 2 × 2 and stride = 2. The LSC of $VGG_{LS}$ performs three max-pooling operations at the output of the first block.

Block	Output	VGG	Filters	${\mathrm{VGG}}_{\mathrm{LS}}$	Filters
1	224 × 224	3 × 3 conv × 2	64	3 × 3 conv × 2	64
	112 × 112	max-pool	64	max-pool	64
2	112 × 112	3 × 3 conv × 2	128	3 × 3 conv × 2	128
	56 × 56	max-pool	128	max-pool	128
3	56 × 56	3 × 3 conv × 2	256	3 × 3 conv × 2	256
	56 × 56	1 × 1 conv	256	1 × 1 conv	256
	28 × 28	max-pool	256	max-pool	256
4	28 × 28	3 × 3 conv × 2	512	3 × 3 conv × 2	512
	28 × 28	1 × 1 conv	512	1 × 1 conv	512
	14 × 14	max-pool	512	max-pool	512
LS	14 × 14			max-pool concat	64
5	14 × 14	3 × 3 conv × 2	512	3 × 3 conv × 2	512
	14 × 14	1 × 1 conv	512	1 × 1 conv	512
	7 × 7	max-pool	512	max-pool	512
classification stage	1 × 1	FC	1	FC	1
	1 × 1	FC	1	FC	1
	1 × 1	soft-max		soft-max	

**Table 2 sensors-23-07582-t002:** Densenet and $Densenet_{LS}$ architecture. The max-pooling layer of the first block is 3 × 3 and stride = 2. Each Dense block is composed of a 1 × 1 convolution layer and a 3 × 3 convolution layer. The transition blocks have a 1 × 1 convolution layer and a 2 × 2 average-pooling layer with stride = 2. The average pool of the first layer and that of the output layer is 7 × 7 with stride = 2. For the LSC of $Densenet_{LS}$, four 3 × 3 max-pooling are performed to reduce the dimension of the first block and concatenate with the fourth block.

Block	Output	Densenet	Filters	${\mathrm{Densenet}}_{\mathrm{LS}}$	Filters
1	112 × 112	7 × 7 conv	48	7 × 7 conv	48
	56 × 56	max-pool	48	max-pool	48
2	56 × 56	dense × 6	192	dense × 6	192
	28 × 28	transition	96	transition	96
3	28 × 28	dense × 12	384	dense × 12	384
	14 × 14	transition	192	transition	192
4	14 × 14	dense × 24	768	dense × 24	768
	7 × 7	transition	384	transition	384
LS	7 × 7			max-pool concat	48
5	7 × 7	dense × 16	768	dense × 16	816
	7 × 7	1 × 1 conv	768	1 × 1 conv	816
classification	1 × 1	ave-pool	1	ave-pool	1
stage	1 × 1	soft-max		soft-max	

**Table 3 sensors-23-07582-t003:** Resnet and $Resnet_{LS}$ architecture. The max-pooling is 3 × 3 with stride = 2, and the average-pooling is 7 × 7 and stride = 2. Each Resnet block is a 1 × 1 conv layer, followed by a 3 × 3 conv layer, and ending with a 1 × 1 conv layer. For the LSC of $Resnet_{LS}$, three 3 × 3 max-pooling are performed to reduce the dimension of the first block and a 1 × 1 conv layer with 1024 filters for adding it to the fourth block output.

Block	Output	Resnet	Filters	${\mathrm{Resnet}}_{\mathrm{LS}}$	Filters
1	112 × 112	7 × 7 conv	64	7 × 7 conv	64
2	56 × 56	max-pool	64	max-pool	64
	56 × 56	resnet × 3	256	resnet × 3	256
3	28 × 28	resnet × 4	512	resnet × 4	512
4	14 × 14	resnet × 6	1024	resnet × 6	1024
LS	14 × 14			max-pool	64
	14 × 14			conv 1 × 1 addition	1024
5	7 × 7	resnet × 3	2048	resnet × 3	2048
classification	1 × 1	ave-pool	1	ave-pool	1
stage	1 × 1	soft-max		soft-max	

**Table 4 sensors-23-07582-t004:** Datasets used in the experiments.

Dataset	Classes	Train/Validation	Theme
Imagenette [25]	10	9469/3925	varied
Tiny Imagenet [26]	200	100,000/10,000	varied
Birds [27]	315	45,995/1590	specialized
Flowers [28]	102	6551/1637	specialized

**Table 5 sensors-23-07582-t005:** Trainable parameters per model and dataset.

	VGG		Densenet		Resnet	
Dataset	o	ls	o	ls	o	ls
IM	129,582,922	129,877,834	3,873,994	3,948,202	23,665,802	24,256,650
TI	130,361,352	130,656,264	4,020,104	4,103,432	27,168,072	27,758,920
BI	130,832,507	131,127,419	4,108,539	4,197,387	29,287,867	29,878,715
FL	129,959,846	130,254,758	3,944,742	4,023,366	25,361,638	25,952,486

IM: Imagenette; TI: Tiny Imagenet; BI: Birds; and FL: Flowers.

**Table 6 sensors-23-07582-t006:** Accuracy obtained by the $M_{O}$ and $M_{LS}$ architectures in the four datasets and absolute (abs.) and relative (rel.) differences between them.

	VGG				Densenet				Resnet			
	Accuracy		Difference		Accuracy		Difference		Accuracy		Difference	
Dataset	o	ls	abs.	rel.	o	ls	abs.	rel.	o	ls	abs.	rel.
IM	64%	74%	10%	14%	**68**%	**75**%	7%	9%	66%	70%	4%	6%
TI	20%	23%	3%	13%	**28**%	**29**%	1%	3%	24%	27%	3%	11%
BI	47%	62%	15%	24%	60%	**77**%	17%	22%	**65**%	74%	9%	12%
FL	25%	43%	18%	42%	**48**%	**57**%	9%	16%	45%	**57**%	12%	21%

IM: Imagenette; TI: Tiny Imagenet; BI: Birds; and FL: Flowers.

**Table 7 sensors-23-07582-t007:** Average training time per epoch (in seconds).

	VGG		Densenet		Resnet	
Dataset	o	ls	o	ls	o	ls
IM	40.137	41.138	33.011	33.051	32.111	33.110
TI	388.124	393.126	314.100	324.104	280.090	284.091
BI	185.128	190.132	140.098	145.101	123.085	128.089
FL	28.129	29.014	22.005	23.113	20.097	21.099

IM: Imagenette; TI: Tiny Imagenet; BI: Birds; and FL: Flowers.

**Table 8 sensors-23-07582-t008:** Pearson correlation between the percentage of filters per CSI interval and the accuracy variation by ablation of filters.

	$M_{O}$	$M_{\mathrm{LS}}$
VGG.	0.009	0.0582
Densenet	−0.0933	−0.0940
Resnet	0.3465	0.1276

**Table 9 sensors-23-07582-t009:** Pearson correlation, for each dataset, between the hue distribution of pixels in the dataset and the number of filters in the model trained with the same dataset.

	VGG16		Densenet		Resnet	
	$M_{O}$	$M_{\mathrm{LS}}$	$M_{O}$	$M_{\mathrm{LS}}$	$M_{O}$	$M_{\mathrm{LS}}$
Imagenette	0.67	0.64	0.85	0.84	0.61	0.60
Tiny Imagenet	0.88	0.87	0.79	0.77	0.75	0.83
Birds	0.75	0.79	0.63	0.71	0.75	0.77
Flowers	0.17	0.32	0.28	0.28	0.20	0.22

## Data Availability

Not applicable.

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
