# Peer review of "A Long Skip Connection for Enhanced Color Selectivity in CNN Architectures"

_sensors, 2023, doi:10.3390/s23177582_

Round 1

Reviewer 1 Report

Your method consists in increasing the color selectivity of a feed-forward neural network by modifying its structure through the incorporation of a long skip connection (LSC) between the output of the initial block. To evaluate the effectiveness of this approach, you proposed a color selectivity index that enables the comparison of the network's activations in response to color images.

The proposed method is very promising and interesting. The experiments carried out are extensive and clearly prove the effectiveness of using the method with LSC.

Figures are clear and the paper is well-written.

I just don't understand why does LSC bring less improvement when there is a strong contrast, while it work better in the opposite complex case ? Shouldn't color differentiation be easier when the contrast is strong?

I agree with testing the method on other more selective datasets in futur works.

Accept in present form.

Reviewer 2 Report

See attachment

Reviewer 3 Report

This paper considers the architectures of CNNs for image classification and proposes the creation of a long skip connection (LSC) between the first and last blocks in the feature extraction component such that deeper parts of a CNN can directly receive information from its shallower layers. Experimental results show that the proposed LSC improves the accuracy of classification and the color selectivity of the original CNN architectures. In addition, a new color representation procedure is proposed to organize and filter feature maps such that visualization of these maps are more manageable. The paper is generally well written. The proposed approach is original and may have its own merits for improving the accuracy of image classification. However, the following issues need to be addressed before it can be accepted for publication.

1. The authors should clearly summarize the major contributions of the paper in the introduction.

2. Does the proposed approach use RGB or other representations for pixels in color images? The authors should clarify this in Section 2.   

3. In Equation (2), activation values a_ji should be explained more clearly. In other words, how can one determine the activation value of a neuron in practice?

4. The additional LSCs would increase the number of parameters that must be determined during the training process, more computation time thus might be needed for training. The authors need to compare the computation time needed for training the new models with that needed to train the original models in Section 4.

5. Adding LSCs in networks is not a new idea and a number of similar approaches exist. Is it possible for the authors to compare the classification accuracy of the proposed approach with that of other similar methods?

Round 2

Reviewer 3 Report

All issues have been addressed. I have no other concerns and recommend the acceptance of the paper.